

# Using in situ GC-MS for analysis of C$_2$-C$_7$ volatile organic acids in ambient air of a boreal forest site

Heidi Hellén.[1], Simon Schallhart[2], Arnaud P. Praplan[1], Tuukka Petäjä.[2] and Hannele Hakola[1]

[1]Finnish Meterological Institute, P.O. Box 503, 00101 Helsinki, Finland

[2]Department of Physics, P.O. Box 64, 00014 University of Helsinki, Finland

*Correspondence to Heidi Hellén (heidi.hellen@fmi.fi)*

**Abstract.** An in situ method for studying C$_2$-C$_7$ monocarboxylic volatile organic acids (VOAs) in ambient air was developed and evaluated. Samples were collected directly into the cold trap of the thermal desorption unit (TD) and analysed in situ using a gas chromatograph (GC) coupled to a mass spectrometer (MS). A polyethylene glycol column was used for separating the acids. The method was validated in the laboratory and tested on the ambient air of a boreal forest in June 2015. Recoveries of VOAs from fluorinated ethylene propylene (FEP) and heated stainless steel inlets were acceptable. Different VOAs were fully desorbed from the cold trap and well separated in the chromatograms. Detection limits varied between 1 and 130 pptv and total uncertainty of the method at mean ambient mixing ratios ranged between 16-76%. All straight chain VOAs except heptanoic acid in the ambient air measurement were found with mixing ratios above the detection limits. The highest mixing ratios were found for acetic acid and the highest relative variations for hexanoic acid. In addition, mixing ratios of acetic and propanoic acids measured by the novel GC-MS method were compared with proton-mass-transfer time-of-flight mass spectrometer (PTR-TOFMS) data. Both instruments showed similar variations, but differences in the mixing ratio levels were significant.





## 1 Introduction

Organic acids comprise a large fraction (~25%) of the non-methane hydrocarbons in the atmosphere (Khare et al., 1999). They are known to have both anthropogenic and biogenic sources (Mellouki et al., 2015). In addition, they are produced in ambient atmospheric air from the oxidation of other

volatile organic compounds (VOCs) (Orzechowska et al., 2005). These reactions include the reaction of ozone with olefinic hydrocarbons, carbonyl oxidation by hydroxyl radicals, and radical recombination reactions between acetyl peroxy and other peroxy radicals (Rosado-Reyes and Francisco, 2006). In addition, anaerobic processes such as composting are well-known sources of volatile organic acids (VOAs) (Brinton, 1998). Acids are usually metabolic by-products of anaerobic

respiration and are breakdown products of more complex organic compounds such as oils and fats present in raw waste. Several VOAs have been found to have high odour potentials at concentrations as low as the ppb level (Brinton, 1998).

The VOAs react with hydroxyl radicals in the air or undergo dry or wet deposition. Aqueous phase reactions provide a sink for water soluble VOAs, but reactions of other VOCs may also be a source

of VOAs (Ervens et al., 2013). The VOAs potentially play a significant role in the production of secondary organic aerosols (Carlton et al., 2006; Zhang et al., 2004). Acids act as an organic coating of aerosol particles (Russell et al., 2002) and they also undergo heterogeneous reactions on particles (Shen et al., 2013; Tong et al., 2010). Heterogeneous reactions of other organic compounds can also produce organic acids. However, VOAs are expected to occur mainly in the gas phase (Yatavelli et

al., 2014). Kawamura et al. (2000) found that $C_1$-$C_{10}$ monocarboxylic acids exist mainly in free volatile forms and the particulate phase fraction represented less than 10% of the total organic acids in the air of southern California in October 1984.

There are several studies on the concentrations of gas phase organic acids in ambient air, but these investigations had predominantly focused on formic and acetic acids (Chebbi and Carlier, 1996 and

25 references there in). In addition, terpenoic acids in particles have been studied using liquid chromatographs (LC) with mass spectrometers (Vestenius et al., 2014; Kristensen and Glasius, 2011), higher carboxylic acids in gas phase simultaneously with ultrafine ($\leq$ 50 nm) particles using GC and LC (Parshintsev et al. 2011) and dicarboxylic acids in particles and gas phase using ion chromatographs with mass spectrometers (e.g. Fisseha et al., 2006). However, these methods are

30 labour intensive and their time resolution is low. Veres et al. (2011) used negative-ion proton-transfer chemical ionization mass spectrometry (PTR-MS) with one minute time resolution to study formic, acrylic, methacrylic, propanoic and pyruvic/butanoic acids in the urban air masses in Pasadena, CA.



In addition a novel online system, filter inlet for gas and aerosols (FIGAERO), has been used with a high resolution time-of-flight chemical ionization mass spectrometer for measurements of formic and monoterpenoic acids in boreal forest (Lopez-Hilfiger et al., 2014).

There are also studies on anthropogenic sources of VOAs: two such studies, Zahn et al. (1997) and

McGinn et al. (2003) measured emissions from pig and beef production facilities. In their studies acids were collected on sorbtion tubes and analysed later by GCs. However, the detection limits for these methods were too high for ambient air studies.

There is a paucity of knowledge of VOAs, other than formic and acetic acids in gas phase, and this dearth of information is at least partly due to the lack of sensitive enough measurement methods for

detecting concentrations in ambient air. In the present study we developed an in situ GC-MS measurement method for measuring $C_2$-$C_7$ monocarboxylic VOAs with two hour time resolution at ambient air concentration levels, which we used to measure ambient air concentrations in a boreal forest site.

**2 Experimental**

**2.1 GC-MS sampling and analysis**

A method for measurements of VOAs in air was developed for an in-situ thermal desorption unit

(Unity 2 + Air Server 2, Markes International ltd.) with a gas chromatograph (Agilent7890) and a mass spectrometer (Agilent 5975C). Samples were taken every other hour. The sampling time was 60 min and the flow 30 ml min$^{-1}$. In the 3 m long fluorinated ethylene propylene (FEP) inlet (1/8 inch I.D.) an extra flow of 2.2 L min$^{-1}$ was used to avoid losses of the compounds on the walls of the inlet tube. Samples were collected directly into the cold trap (U-T17O3P-2S, Markes International Ltd.)

of the thermal desorbtion unit. All the lines and valves in the thermal desorbtion unit were kept at 200ºC. Water was removed by keeping the hydrophobic cold trap at 25 ºC during sampling and using a post sampling line purge (10 minutes, 30 ml/min), post sampling trap purge (10 minutes, 20 ml/min) and pre-trap fire purge (10 minutes, 10-11 ml/min). For desorption the cold trap was heated to 300ºC for 3 minutes and flushed with a helium flow of 10-11 ml min$^{-1}$. The poluethylene glycol column

used for separation was the 30-m-long DB-WAXetr (J&W 122-7332, Agilent) with an inner diameter of 0.25 mm and a film thickness of 0.25 µm. Helium (99.9996%) was used as a carrier gas. During the analysis the GC oven was first kept at 50 ºC for 10 min, heated to 150 ºC with the rate of 4 ºC



min$^{-1}$ and then to 250ºC with the rate of 8 ºC min$^{-1}$, where it was kept for 5 min. The total run time was 52.5 min.

The system was calibrated using liquid standards in Milli-Q water injected into adsorbent tubes filled with Tenax TA and Carbopack B. After injection the tubes were flushed with nitrogen (N$_2$, 99.9999%) flow of 80 ml/min for 10 minutes to remove the water. Standard tubes were desorbed and analysed using the same method as for the samples. Fresh standards were prepared from a volatile free acid mixture (CRM46975, Supelco) one day before the analysis. The stability of the mass spectrometer was followed by running gaseous field standards containing aldehydes and aromatic hydrocarbons

after every 50[th] sample taken and using tetrachloromethane as an 'internal standard'. The concentration of tetrachloromethane in ambient air is stable, thus it was possible to detect sampling errors or shifts in calibration levels by following its concentration.

## 2.2 Test site and ambient air measurements

An ambient air sampling campaign was conducted at SMEAR II forest research station in Hyytiälä (61º51'N, 24º17'E, 181 m a.s.l), Finland, between 11 and 27 June 2015. The SMEAR II station is a dedicated facility for studies of forest ecosystem-atmosphere associations (Hari and Kulmala, 2005). The measurement station is located in a Scots pine stand that is approximately 50-years-old. The

continuous measurements at that location include leaf, stand and ecosystem-scale measurements of greenhouse gases, VOCs, pollutants (e.g. O$_3$, SO$_2$, NO$_x$) and many different aerosol properties. In addition, a full suite of meteorological measurements of the site is continuously recording.

## 2.3 PTR-TOFMS measurements

During the measurement campaign at SMEAR II a PTR-TOFMS (Ionicon Analytik GmbH; Graus et al., 2010, Jordan et al., 2009) was run in parallel with in situ GC-MS. The PTR-TOFMS instrument was operated at a drift tube pressure of 2.3 mbar and a drift tube voltage of 600V. These settings resulted in an *E/N* of 130 Td, where *E* is the electrical field strength and *N* the gas number density. The air was sampled at a flow of 20 l min$^{-1}$ through a 3.5 m PTFE inlet, which had an inner diameter

of 4 mm. A total flow at the rate of 500 ml min$^{-1}$ went to the instrument via a three way valve (type: 6606 with ETFE, Bürkert GmbH & Co. KG), 10 cm of 1.6 mm (I.D.) PTFE and 10 cm of 1mm (I.D.) PEEK tubing. There, 30 ml min$^{-1}$ of the flow was sampled and the remainder served only as a by-pass flow in order to decrease the response time and wall losses. A 20 min background measurement was



performed three times a day, during which the air from the 3.5 m inlet was let through a custom build catalytic converter. The second port of the three way valve was used for this. The instrument was calibrated every 2-3 weeks, as described in Schallhart et al. (2016). The calibration gas did not contain acetic acid or propanoic acid, and their fragmentation pattern was not quantified, the sensitivity was

estimated as being 50% of the acetone sensitivity.The instrumental background for acetic acid was clearly correlated with ambient measurements. This can be explained by a memory effect (of the inlet and/or instrument) of those compounds. This has already been observed by de Gouw et al. (2003). Therefore, the reported concentrations of acetic acid are underestimated, as an excessively high background signal had been subtracted. The mean detection limits for acetic and propanoic acids

during the campaign were 34 and 8 pptv, respectively.

## 2.4 Calculation of the uncertainties

Total uncertainty of the measurements ($U_{tot}$) was calculated from precision ($U_{prec}$) and systematic

errors ($U_{sys}$):

$$U_{tot}^2 = U_{prec}^2 + U_{sys}^2 \qquad (1)$$

The precision was calculated thus:

$$U_{prec} = \frac{1}{3}DL + RSD \times \chi \qquad (2)$$

Where DL is the detection limit, RSD relative standard deviation between the samples, when known amounts of acids were injected into the $N_2$ flow and $\chi$ is the mean mixing ratio of the acid in ambient

air during the measurement campaign at the SMEAR II site. The detection limit is the dominant factor for low mixing ratios whereas the secondary term used describing reproducibility of the instrument and this becomes more important for higher mixing ratios.

The systematic error includes uncertainty of the standard solution ($U_{stdmix}$) given by the producer, uncertainty of the standard preparation ($U_{stdprep}$) estimated for the equipment that was used,

uncertainty of the sample volume ($U_{vol}$) that was obtained for the uncertainty of the mass flow



controller, errors due to blank corrections ($U_{blank}$) and further instrument problems (e.g. error due to correction of the drift of the calibrations using tetrachloromethane, $U_{drift}$):

$$U_{sys}^2 = U_{stdmix}^2 + U_{stdprep}^2 + U_{vol}^2 + U_{blank}^2 + U_{drift}^2 \qquad (3)$$

## 3 Results

### 3.1 Method validation

Peaks of the different acids were separated very well in the chromatogram (Table 1). Background values of VOAs in the system were estimated by sampling clean nitrogen (HiQ $N_2$ 6.0 >99.9999%)

using the same method as used for the samples. Blank values were obtained for acetic, propanoic and butanoic acids (Table 1). The detection limits were defined as three times the standard deviation of the blank values or alternatively as signal-to-noise ratio (3:1). Detection limits varied between 1 and 130 pptv and were highest for acetic acid due to the high blank values.

Some memory effect was found for all studied acids after running calibration tubes and standard

gases, but these disappeared after measuring 5 consecutive samples. The calibration standards contained amounts that corresponded to ambient mixing ratios up to 10 000 pptv, whereas the mean ambient mixing ratios varied between 1 and 1160 pptv. Therefore, the first five samples after calibrations were always disregarded. Using lower concentration for the calibrations would be expected to solve this issue.

The desorption efficiency (DE) of the cold trap was determined by redesorption at a higher temperature (320 °C) after running a sample. The amount of the sample found in the first desorption was compared to the total amount of the sample. DEs from the cold trap were >98% for all compounds.

The precision ($U_{prec}$) was checked by injecting known amounts of acids into the $N_2$ flow. Mixing

ratios varied between 0.1 and 1994 ppbv. The precision calculated for the ambient mixing ratios found at SMEAR II using the Eq. (2) was found to vary from 7 to 38% for the acids of interest. The total expanded uncertainties of the studied acids varied between 16 and 76% (Table 1). The highest relative uncertainties were found for the compounds with mixing ratios closest to the detection limits. The uncertainties for benzene and toluene were as high as 108 and 72%, respectively. Earlier studies that



used the same instrument (Kajos et al., 2015) found the relative analytical uncertainties of benzene and toluene to be much lower (4 and 5%, respectively). However, the present study found the mean mixing ratio of benzene was at the detection limit (20 pptv) and the mean mixing ratio of toluene was very close to it. The uncertainties are given for these low values, thus they are expected to be much

higher.

The real uncertainty of the acetic acid in these measurements is expected to be higher than that reported due to calibration issues mentioned above. The precision for the acetic acid was good (7%), but acetic acid has an additional systematic error, which was not found for the other compounds studied. There was a high background level of acetic acid in the calibrations, which was probably due

to the preparation of the calibration solutions and adsorbent tube standards that caused non-linearity of the calibration curve. This high background concentration was estimated by analyzing blank adsorbent tubes, i.e. tubes that had been prepared with only the solvents but without any acetic acid. A better calibration method such as one that uses the permeation device could remove this source of uncertainty.

It is expected that a proportion of acids will be lost in the inlet tubes, therefore inlet loss estimation tests were conducted using a permeation oven (FlexStream Base, Kin-Tek) with a nitrogen flow of 0.50 or 0.75 l min$^{-1}$. The permeation vials were filled with the studied acids and placed into the oven at 40ºC. These tests were performed both with dry and humid nitrogen flow and the concentrations of acids varied between 0.2 and 1994 ppb (Table 2). Four different configurations were tested: 1) One

with humidified N$_2$ flow of 0.75 l min$^{-1}$ and 4 m long FEP tube (i.d. 1/8 inch) at room temperature, 2) one with humidified N$_2$ flow of 0.75 l min$^{-1}$ and 1 m long stainless steel tube (i.d. 0.069 inch) heated to 120ºC and used for ozone removal in terpenoid sampling (Hellén et al., 2012), 3) one with humidified N$_2$ flow of 0.75 l min$^{-1}$ and 3 m long FEP tube heated to 120ºC, and 4) one with dry N$_2$ flow of 0.50 l min$^{-1}$ and 3 m long FEP tube heated to 120ºC. Samples were taken before and after the

inlets. The comparison results for toluene are included in Table 2. The results for all configurations were acceptable (within ±20%). The first configuration was chosen for further tests and for ambient air sampling. The ozone removal tube was not selected because the studied acids are not reactive towards ozone, but the test was conducted for the situations where ozone reactive compounds (e.g. sesquiterpenes) can be measured using the same system.





### 3.2 Results from ambient air measurements

### 3.2.1 Mixing ratios in a boreal forest

The highest mixing ratios were measured for acetic acid (Table 3). The mixing ratios of isobutyric, isohexanoic and heptanoic acids stayed below their detection limits during the whole campaign. The
mixing ratios generally decreased with increasing carbon number except for hexanoic acid. Hexanoic acid was more abundant than pentanoic acid. Such a VOA profile was also seen in the measurements of Kawamura et al. (2000) but in the urban air of southern California in 1984.

Hexanoic acid had the highest relative variations in mixing ratios (Fig. 1). The variation in sources and source strenghts together with higher reactivity of hexanoic acid may explain this. Reaction rates
of VOAs with hydroxyl radicals increased with increasing carbon number (Mellouki et al., 2015) and trees and other vegetation are known to produce stress induced emissions of green leave volatile organic compounds which are aldehydes, esters and alcohols with 6-carbon atoms (Hakola et al., 2001; Scala et al., 2013). Oxidation of these compounds could be a source of hexanoic acid. However, based on the current knowledge even direct emissions of hexanoic acid cannot be ruled out.

Butanoic acid emissions peaked (100 pptv) on 14 June (Fig. 1). This peak occurred at the same time asthe peak of 1-butanol (2500 pptv). 1-Butanol was being used at the same site in other instruments including particle counters. During malfunctions of these instruments 1-butanol may have been released into the ambient air. Butanoic acid was expected to be produced in the oxidation reactions of 1-butanol in the atmosphere. Maximum mixing ratio occurred in the middle of the night (1:30-2:30
AM, local time), which gave an indication that butanoic acid has been produced from nitrate radical reactions.

Acetic acid was measured at the same site in August 2001 using an annular denuder system and IC analysis (Boy et al., 2004). The diurnal means of concentrations of acetic acid varied between 166 and 1666 pptv, which is close to values measured in this present study in June 2015. Information on
mixing ratios of VOAs higher than $C_2$ is scarce. Kawamura et al. (2000) measured $C_1$-$C_{10}$ VOAs in southern California in October 1984 and their mixing ratios were at similar levels as found in our measurements in the present study (Table 3). However, Veres et al. (2011) found clearly higher mixing ratios using negative-ion proton transfer chemical-ionization mass spectrometry in June 2010 in Pasadena California. The mean mixing ratio of propionic acid was 1740 pptv whereas it was only
81 pptv in our study and 29-211 pptv in the study of Kawamura et al. (2000). Veres et al. (2011) found evidence that organic acids were photochemically and rapidly produced from urban emissions





transported from Los Angeles. Nolte et al. (1999) also detected much higher mixing ratios of $C_2$-$C_{10}$ acids at the four urban sites in Southern California in Sepetmber 1993, but mixing ratios found at San Nicolas Island (background) were lower than in our measurements. The vegetation in Southern California is very different compared to our boreal site and differences in primary and secondary sources may explain the differences.

### 3.2.2 Diurnal variation of mixing ratios

Acetic and propanoic acids had the highest mixing ratios during the day and lowest during the night (Fig. 2). Hexanoic acid had the opposite diurnal variation with the maximum concetration occurring during the night. Butanoic and pentanoic acids did not show any clear diurnal cycle. Direct emissions from vegetation and production in photochemical reactions are expected to be highest during the day when there is more light and higher temperature. However, reactions of VOAs and mixing are also faster during the day and this phenomenon, in addition to the lower boundary layer present during the night, may explain the high night-time concentrations of faster reacting VOAs. High night-time concentrations have also been measured at the site for monoterpenes even though their emissions are clearly highest during the day (Hakola et al. 2012). During the night VOAs may also be produced from ozone and nitrate radical reactions.

Similar diurnal pattern of propionic acid with daytime maxima was also found in the study of Veres et al. (2011) in California in June 2010. Those authors found daytime maxima for pyruvic/butanoic acid, but in the present study we found that butanoic acid did not have any clear diurnal variation.

### 3.2.3 Comparison with other trace gases and meteorological parameters.

Data for the other trace gases and meteorological parameters (Fig. 1) were obtained from the SmartSmear AVAA portal (Junninen et al. 2009, Williams et al., 2011). All data is for the height of 4.2 m except wind speed, which is for 8.4 m. Acetic acid had a weak correlation with temperature ($R^2$=0.35) and propanoic acid with ozone ($R^2$=0.25). Hexanoic acid concentration correlated with toluene ($R^2$=0.42), α-pinene ($R^2$=0.42), and CO ($R^2$=0.52). The highest hexanoic acid concentrations were measured during nights with low wind speed. This indicates that mixing ratios of shorter chain VOAs were more dependent on photochemical production or temperature and light dependent emissions, whereas the diurnal cycle of longer chain VOAs were more strongly affected by reactivity and mixing of air.



### 3.2.4 Comparison with PTR-TOFMS data

The PTR-TOFMS measured acetic and propanoic acids, whereas the other VOAs remained below their respective detection limits. The variations of the mixing ratios were quite similar for both instruments (Fig. 3). The correlation was realtively good when the mixing ratios of acetic acid (GC > 1300 ppt, $R^2$=0.78) and propanoic acid (GC > 80 ppt, $R^2$=0.52) were highest. Low correlations with lower values were expected due to the high uncertainties for both instruments when the levels of the VOAs being analysed were close to their respective detection limits.

The mean mixing ratios of acetic and propanoic acids measured by GC-MS were 5.7 and 2.3 higher than those measured by the PTR-TOFMS method. The main reason for the large discrepancy for acetic acid is the overestimation of the background due to memory effects in the PTR-TOFMS as discussed in section 2.3. The measurements were conducted in separate containers, but were close to each other (5m). Therefore, some differences were expected, but not large differences. The overall variations of the signal of the two instruments are compareable, thus the main difference between them seems to be due to the background problem or problems in calibrations of the instruments. The calibration curve of acetic acid for the GC-MS measurements suffered from high background at low levels. More accurate measurements of these compounds require that better calibration methods are developed. In addition to this, using different inlet line and valve materials could help to reduce the memory effect and lower the backgound.

## 3   Conclusions

A novel in situ GC-MS method for the quantification of volatile organic acids was evaluated. Despite the relatively high uncertainty, the method is uniquely capable of detecting VOAs at low concentrations with only a 2-hour-time resolution. Experimentally determined recoveries of VOAs from FEP and heated stainless steel inlets were acceptable and different VOAs were fully desorbed from the cold trap and were well separated in the chromatograms. Detection limits varied between 1 and 130 pptv between individual VOAs.

The mixing ratios of acetic and propanoic acids measured with the novel GC-MS method were compared to PTR-TOFMS data. Similar variations of mixing ratios were captured by both analytical set-ups, but absolute levels deviated significantly. High background concentration was a problem for both instruments and especially for the measurement of acetic acid by the PTR-TOFMS method. Replacing the inlet line and valve materials could improve the situation. A better calibration method



especially for acetic acid especially in GC-MS measurements, would also improve the quality of the data for acetic acid.

The system performed well for ambient air measurements at a boreal forest site. We found that acetic acid had the highest mixing ratios, but hexanoic acid concentrations varied the most. The lightest VOAs (acetic and propanoic acids) had their maxima in the afternoon, whereas hexanoic acid had opposite diurnal variation.

This novel in situ TD-GC-MS method will allow us to study diurnal and seasonal variations of VOAs in ambient air and produce new data on, which will benefit atmospheric chemistry and new particle formation studies.

**Acknowledgements**

The research was supported by the Academy research fellow project (Academy of Finland, project 275608) and Academy of Finland via Center of Excellence in Atmospheric Sciences. Data providers of SmartSmear AVAA portal are gratefully acknowledged.

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



**Table 1.** Retention times (RT), blank values (BL), detection limits (DL), precision (U_Prec) and total expanded uncertainties (U) for studied compounds at mean ambient air mixing ratios during the measurement campaign at SMEAR II in June 2015.

|  | RT (min) | BL (pptv) | DL (pptv) | $U_{prec}$ (%) | $U_{tot}$(%) |
|---|---|---|---|---|---|
| Acetic acid | 31.3 | 156 | 130 | 7 | 16* |
| Propanoic acid | 34.4 | 5 | 23 | 15 | 32 |
| Isobutyric acid | 35.4 | - | 16 | - | - |
| Butanoic acid | 37.3 | 3 | 7 | 19 | 39 |
| Isopentanoic acid | 38.3 | - | 1 | - | - |
| Pentanoic acid | 40.0 | - | 5 | 38 | 76 |
| Isohexanoic acid | 41.6 | - | 13 | - | - |
| Hexanoic acid | 42.5 | - | 7 | 20 | 40 |
| Heptanoic acid | 44.7 | - | 19 | - | - |
| Benzene | 8.4 | 6 | 20 | 53 | 108 |
| Toluene | 12.4 | 8 | 9 | 35 | 72 |

*Acetic acid has an additional error source which was not taken into account in these calculations (see main text).



**Table 2.** Recoveries (%) from the inlets together with amounts and mixing ratios (vmr) used in the tests.

|  | amount | vmr | 1 | 2 | 3 | 4 |
|---|---|---|---|---|---|---|
|  | ng sample$^{-1}$ | ppbv | % | % | % | % |
| Acetic acid | 8.6 | 4.0 | 101 | 104 | 98 | 97 |
| Propanoic acid | 1.7 | 0.6 | 105 | 107 | 109 | - |
| Isobutyric acid | 6470 | 1992 | 99 | 100 | 112 | 90 |
| Butanoic acid | 109 | 16 | 96 | 101 | 108 | 95 |
| Pentanoic acid | 0.8 | 0.2 | 87 | 98 | 123 | 94 |
| Hexanoic acid | 16 | 3.6 | 104 | 107 | 93 | 98 |
| Toluene | 15 | 4.6 | 100 | 101 | 105 | 97 |

1) 4 m FEP tube (i.d. 1/8 inch) at room temp, humidified $N_2$ flow 0.75 l min$^{-1}$
2) 1 m stainless steel tube (i.d. 0.069 inch) at 120ºC, humidified $N_2$ flow 0.75 l min$^{-1}$
3) 3 m FEP tube (i.d. 1/16 inch) at 120ºC, humidified $N_2$ flow 0.75 l min$^{-1}$
4) 3 m FEP tube (i.d. 1/16 inch) at 120ºC, dry $N_2$ flow 0.75 l min$^{-1}$

25



**Table 3.** Mixing ratios (pptv) of volatile organic acids at SMEAR II station in Hyytiälä, Finland, between 11 and 27 June 2015 and in earlier studies.

| | Present study | | | Nolte et al. (1999) | | Kawamura et al. (2000) | Veres et al. (2011) |
|---|---|---|---|---|---|---|---|
| pptv | Mean | Min | Max | Background | Urban | Urban | Rural |
| Acetic acid | 1160 | 910 | 1520 | 720 | 6560 | 290-2640 | - |
| Propanoic acid | 81 | <DL | 130 | 30 | 550 | 29-211 | 0-6100 |
| Isobutyric acid | <DL | <DL | 20 | 6 | 80 | 5-18 | - |
| Butanoic acid | 40 | 20 | 100 | 3 | 160 | 9-50 | 0-240 |
| Isopentanoic acid | 1 | <DL | 4 | - | - | - | - |
| Pentanoic acid | 10 | <DL | 20 | 0 | 60 | 3-20 | - |
| Isohexanoic acid | <DL | <DL | <DL | - | - | - | - |
| Hexanoic acid | 20 | <DL | 80 | 4 | 90 | 4-32 | - |
| Heptanoic acid | <DL | <DL | <DL | 0 | 30 | 2-30 | - |
| Benzene | 20 | <DL | 90 | | | - | - |
| Toluene | 20 | <DL | 70 | | | - | - |





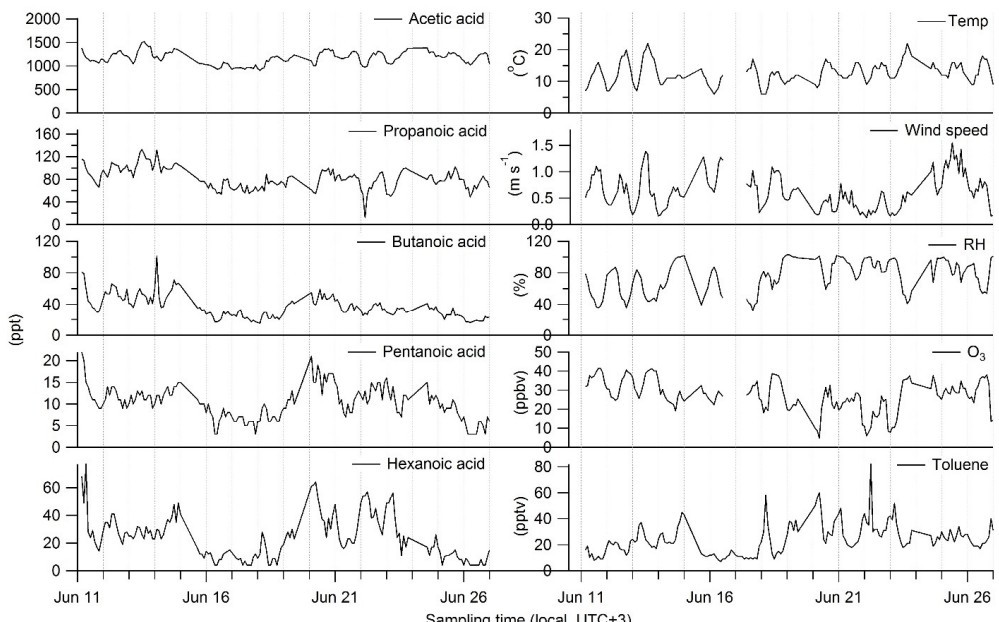

**Fig. 1.** Mixing ratios (pptv) of volatile organic acids ($C_2$-$C_6$) and trace gases together with meteorological parameters at SMEAR II station in Hyytiälä, Finland.





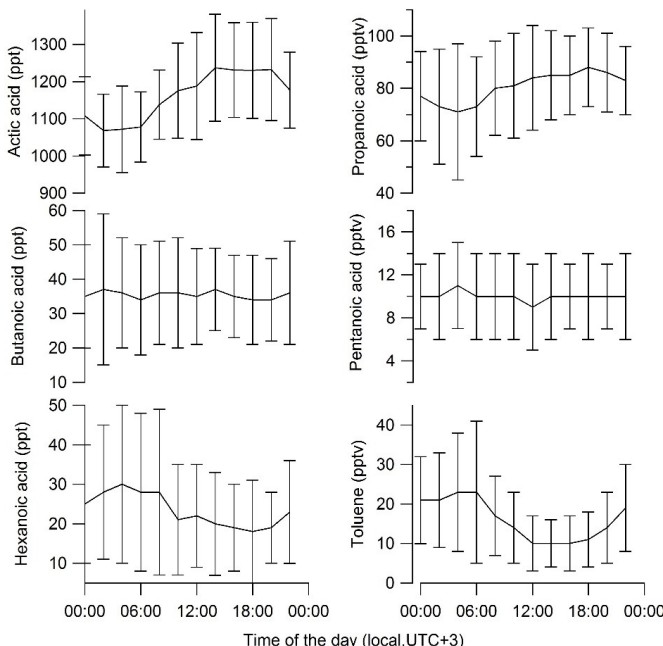

**Fig. 2.** Mean diurnal variation of the mixing ratios with standard devaitions (error bars) at SMEAR II between 11 and 27 June 2015.





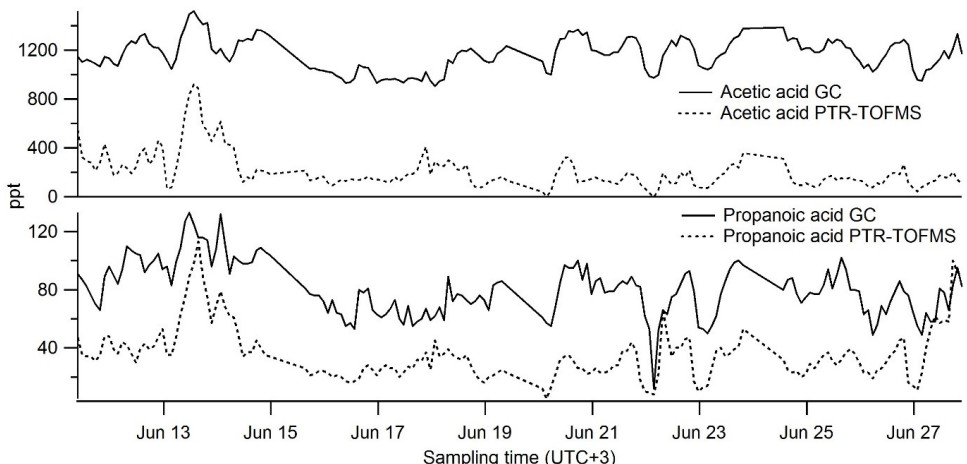

**Fig. 3.** Comparison of mixing ratios (pptv) measured by GC-MS and PTR-TOFMS at SMEAR II in June 2015.