# Peer review of "Using in situ GC-MS for analysis of C2-C7 volatile organic acids in ambient air of a boreal forest site"

_Atmospheric Measurement Techniques, 2016_

## Referee Comment (RC1) · Anonymous Referee #1 · 21 Oct 2016

The authors present the development of an analytical method for the analysis of volatile organic acids (VOA) by a cryotrapping approach and in situ GC-MS analysis. Further, the method was applied to ambient air of a boreal forest measurement site in Hyytiälä, Finland, which can be regarded as a background site for biogenic emissions with low anthropogenic influence, thus presenting an analytical challenge in terms of limits of detection and limits of quantification in the analysis of VOA. For nine linear and branched aliphatic carboxylic acids the method was thoroughly developed, including a dedicated discussion and calculation of uncertainties and tests for inlet losses under different conditions. The manuscript is well suited for publication in AMT, because the presented analytical method allows for the quantification of VOA at ppt concentration levels, which are relevant for VOA other than acetic and formic acid in the ambient atmosphere, with a relatively high time resolution. However, organization and presentation of the paper should be improved prior to publication as outlined in the specific comments below.

Specific comments:

1. The introduction could be improved by adding additional information on the relevance of VOA, especially of the investigated aliphatic monocarboxylic acids, in the atmosphere.

2. The discussion of the reported uncertainties of benzene and toluene in section 3.1 could be clarified, e.g., by referencing equation (2) and explaining the influence of measurements close to the DL on the respective uncertainty.

3. The authors tested losses of VOA in their inlet tubing. Could you please comment on memory effects of VOA other than acetic acid, which may be due to the evaporation of adsorbed VOA in the inlet tubes? Would this play a role over the timescale of sampling?

4. In my opinion the paper would benefit from showing an exemplary GC-MS chromatogram, at least in a separate supporting information file. Which (pairs of) ions were used as mass traces for the respective VOA? This information could be included in Table 1.

5. Calibration curves could also be included in a supporting information file, to show more of the analytical performance of the developed method.

6. The discussion of diurnal patterns of VOA mixing ratios in section 3.2.2 should be backed up by more references, e.g., on P. 9 L. 16: please provide references for the production of VOA from ozone and nitrate radical reactions.

7. Correlation plots of VOA vs. other trace gases and meteorological parameters discussed in section 3.2.3 should either be included in the main manuscript or in a respective supporting information file.

Exemplary technical comments:

P.1, L. 18 "were acceptable" – please be more specific here

P. 5, L. 2: "The second port of the three-way valve was used for this" - this sentence could be removed

P. 11, L. 1: Remove the second "especially"

P. 19, Fig. 2: "devaitions" should be "deviations"

---

## Referee Comment (RC2) · Anonymous Referee #2 · 14 Nov 2016

The authors present the use/development of an in situ GC-MS technique to measure ambient volatile organic acids (VOAs). The method is based on a cold trap technique followed by GC-MS analysis. The method was tested and validated on standard acids. Although the technique used in this study seems to lead to the analysis at "pptv" levels of C2-C7 monocarboxylic acids in the ambient atmosphere, the paper could benefits from a major revision mainly to improve the organization, editing as well as scientific discussion. The introduction could be improved if the authors incorporate a concise idea about VOAs in ambient atmosphere (origins, gas phase VOAs, particle phase VOAs, urban and rural VOAs). In many places throw-out the manuscript, some ideas were repeated or confusing (see my comments below). The authors are encouraged to incorporate in the introduction references and studies associated with VOAs in ambient atmosphere in a concise way. I was confused when the authors referring to

VOAs in ambient air! It will be beneficial for the readers if the authors discuss the utility, importance and use of the technique for gas and aerosol samples? It was unclear in the text if both ambient gas and aerosol were sampled together? Although as noted by the authors, most VOAs may exist in the gas phase but due to partition and the presence of polar acidic group in these compounds, they may also be in the particle phase (aerosol). The sampling site is dominated by biogenic emissions including isoprene and monoterpenes (biogenics), and it will be great if the authors provide chromatograms associated with test samples and ambient samples in the manuscript. Are there other acids in the chromatogram? e.g. methylglyceric acid and other acids (peaks) from isoprene/other HCs oxidation? Is the technique can be used for dicarboxylic acids?

Specific comments: 1. Line 2, page 2. Is "Organic acids" comprise $\sim$25% of the non-methane hydrocarbons in the atmosphere? It's very high to me! It's true that 25% of the non-methane hydrocarbon in the atmosphere are organic acids (including gas phase and aerosol phase). Please check for accuracy! Is organic acids referred here are only compounds having carboxylic acid groups or with multifunctional groups including at least one acid group!!! 2. "The VOAs react with hydroxyl radicals in the air or undergo dry or wet deposition." Needs reference(s)?? 3. Not sure what the authors refer in this sentence "Aqueous phase reactions provide a sink for water soluble VOAs, but reactions of other VOCs may also be a source of VOAs (Ervens et al., 2013)." 4. Additional information of VOAs is suitable in the introduction mainly acids studied in this study. 5. The authors refer the role of acids in SOA formation, are the acids studied here relevant to SOA formation? Please elaborate? 6. It will be beneficial to clarify if gas phase or aerosol phase VOAs or both were measured here? 7. The discussion related to GC-MS sampling and analysis could be clarified. Confused about the sampling directly to the cold trap? Using test tubes etc.. How the samples were taken from the field site!! 6. The paper could be improved significantly if the sampling method was clearly stated. A schematic diagram may be beneficial? 7. Is the PTR-TOFMS discussion necessary in this study? 8. What the authors refer here "The

second port of the three way valve was used for this." Page 5, line 2. 9. Please elaborate in "and their fragmentation pattern was not quantified" page 5, line 4 10. How the memory effect can be incorporated in this study? It does depends on the acids concentrations in the previous runs? Please elaborate? How it was different from one acid to the other? 11. Confused about this statement "Using lower concentration for the calibrations would be expected to solve this issue." page 5, line 19? It did or did not solve the problem? 12. Which N2 flow the authors refers here?? "The precision (Uprec) was checked by injecting known amounts of acids into the N2 flow."?

The method developed here is of great interest, however at this stage, I believe a major work need to be done before its acceptance.
* * *

---

## Referee Comment (RC3) · Anonymous Referee #3 · 16 Nov 2016

The paper describes the development and application of an in situ method for the measurement of C2-C7 monocarboxylic acids (VOAs). The method is based on direct sampling into a commercial cold trap of a thermal desorption unit followed by GC-MS. The authors present results of laboratory experiments to validate the method and apply the method for a series of ambient air samples (boreal forest, June 2015). One focus of the manuscript is for example the investigation of the recoveries of the analytes from FEP and stainless steel inlets. In conclusion the manuscript reports detection limits between 1 and 130 pptv and a total uncertainty of the concentration measurements of about 16–76 percent. Finally, the authors compare the results for selected analytes measured by the GC-MS method PTR-TOF-MS measurements and observe large discrepancies between both techniques. The quantitative determination of organic compounds in ambient air is still a challenging task, especially if a higher time resolution is

required. Therefore, the topic of the manuscript is well suited to be published in AMT. However, several parts of the manuscript should be improved before final acceptance.

One of my major concern is the question how quantitative is the presented method for the target analytes. The authors report by themselves that especially acetic acid showed a problematic behavior and that large deviations were observed. Just considering the sampling step in the "cold" trap at 25°C (to avoid water interference) in combination with the calibration using liquid standards injected into adsorbent tubes (purged with nitrogen – again to avoid water interference) results in the inherent difficulty to estimate a reliable recovery, since for both steps losses of the analyte (e.g. esspercially acetic acid) cannot be excluded (or better have to be expected!?). If no additional experiments can be performed, at least a comprehensive discussion about such losses are needed, which would certainly improve the manuscript.

Specific remarks:

Page 6: "Some memory effect was found . . .". Please describe more quantitative these effects since especially for the highly polar and sticky analyte molecules such a behavior has to be known in detail.

Page 8, line 20: As also discussed later in the manuscript: High concentrations during nighttime are not necessarily a consequence of nitrate chemistry (e.g. transport).

Please check the whole text for typos (using a spell checker!?).
* * *

---

## Author Comment (AC1) · 23 Nov 2016

Thank you for the very good comments. We have carefully considered all the comments and improved our manuscript according to them. Please, find below our answers to the specific and technical comments.

Specific comments: 1. The introduction could be improved by adding additional information on the rel- evance of VOA, especially of the investigated aliphatic monocarboxylic acids, in the atmosphere.

-Introduction was improved by adding more discussion on the sources and fate of studied VOAs in the atmosphere

2. The discussion of the reported uncertainties of benzene and toluene in section

3.1 could be clarified, e.g., by referencing equation (2) and explaining the influence of measurements close to the DL on the respective uncertainty.

-Reference to Eq.(2) and better explanation was added to the manuscript in section 3.1

3. The authors tested losses of VOA in their inlet tubing. Could you please comment on memory effects of VOA other than acetic acid, which may be due to the evaporation of adsorbed VOA in the inlet tubes? Would this play a role over the timescale of sampling?

-There is some memory effect for all studied VOAs. Due to memory effects fast variations of mixing ratios would not be detected. However, this system is relatively slow (60 minute samples every other hour) and unable to detect fast variations anyway. Memory effect was clearly seen after running field standard gas which contained VOAs, but it was only <3% already in the first run after the standard as shown by running blank gas ($N_2$) after the field standard. We deleted always 5 samples after running this standard. In inlet tests we did not detect any major losses of these compounds. Discussion on memory effect was added to the text and figures of blanks and first runs after field standards were added to the supplementary material.

4. In my opinion the paper would benefit from showing an exemplary GC-MS chromatogram, at least in a separate supporting information file. Which (pairs of) ions were used as mass traces for the respective VOA? This information could be included in Table 1.

-Examples of the selected ion chromatograms of calibration and ambient air for m/z 59.9 and 73.9 were added to the supplement and referenced in the section 3.1. In SCAN mode chromatograms background was so high that peaks of studied compounds were not detected. Mass tracer ions were added in the Table 1 and mentioned in the section 2.1.

5. Calibration curves could also be included in a supporting information file, to show more of the analytical performance of the developed method.

-calibration curves for different VOAs were added as supplement Figure S3.

6. The discussion of diurnal patterns of VOA mixing ratios in section 3.2.2 should be backed up by more references, e.g., on P. 9 L. 16: please provide references for the production of VOA from ozone and nitrate radical reactions.

-References were added.

7. Correlation plots of VOA vs. other trace gases and meteorological parameters discussed in section 3.2.3 should either be included in the main manuscript or in a respective supporting information file.

-Correlation plots were added as supplement Figure S6.

Exemplary technical comments: P.1, L. 18 "were acceptable" – please be more specific here

-A range of values was added

P. 5, L. 2: "The second port of the three-way valve was used for this" - this entence could be removed

-This sentence was removed

P. 11, L. 1: Remove the second "especially"

-This was removed

P. 19, Fig. 2: "devaitions" should be "deviations"

-This typo was corrected

---

## Author Comment (AC2) · 23 Nov 2016

Thank you for the good comments. We have carefully considered all the comments and greatly improved our manuscript according to them. Please, find below our answers to the general and specific comments.

The introduction could be improved if the authors incorporate a concise idea about VOAs in ambient atmosphere (origins, gas phase VOAs, particle phase VOAs, urban and rural VOAs). In many places throw-out the manuscript, some ideas were repeated or confusing (see my comments below). The authors are encouraged to incorporate in the introduction references and studies associated with VOAs in ambient atmosphere in a concise way.

-Introduction was improved by adding discussions on sources and fate of the studied

[Figure]

VOAs in the atmosphere and several parts of the manuscript were clarified.

I was confused when the authors referring to VOAs in ambient air! It will be beneficial for the readers if the authors discuss the utility, importance and use of the technique for gas and aerosol samples? It was unclear in the text if both ambient gas and aerosol were sampled together? Althoug has noted by the authors, most VOAs may exist in the gas phase but due to partition and the presence of polar acidic group in these compounds, they may also be in the particle phase (aerosol).

-This methods was suitable only for gas phase samples. Small particles are lost in the FEP inlet lines and bigger ones follow the main gas flow when the sub-sample is taken to the thermal desorption unit. This is clarified in the section 2.1.

The sampling site is dominated by biogenic emissions including isoprene and monoterpenes (biogenics), and it will be great if the authors provide chromatograms associated with test samples and ambient samples in the manuscript. Are there other acids in the chromatogram? e.g. methylglyceric acid and other acids (peaks) from isoprene/other HCs oxidation? Is the technique can be used for dicarboxylic acids?

-Examples of the selected ion chromatograms of calibration and ambient air samples for m/z 59.9 and 73.9 were added as supplements and referenced in the section 3.1. In SCAN mode chromatograms the background was so high that peaks of the studied compounds or any other acids were not detected. This method is suitable only for gas phase samples and most of the dicarboxylicacids are in the particle phase or partitioning between gas and particle phases. Isoprene emissions at this boreal forest site are low and acids produced from mono- or sesquiterpenes are known to be less volatile (main products dicarboxylic acids) and partition between gas and aerosol phases. Therefore we did not try to look for those acids with this system. Other methods would be more suitable for their detection.

Specific comments: 1.Line 2, page 2. Is "Organic acids" comprise 25% of the non-methane hydrocarbons in the atmosphere? It's very high to me! It's true that 25% of the

non-methane hydrocarbon in the atmosphere are organic acids (including gas phase and aerosol phase). Please check for accuracy! Is organic acids referred here are only compounds having carboxylic acid groups or with multifunctional groups including at least one acid group!!!

-Since 25% was not well-defined, we removed it.

2. "The VOAs react with hydroxyl radicals in the air or undergo dry or wet deposition." Needs reference(s)??

-References were added

3. Not sure what the authors refer in this sentence "Aqueous phase reactions provide a sink for water soluble VOAs, but reactions of other VOCs may also be a source of VOAs (Ervens et al., 2013)."

-This was removed, when the introduction was improved

4. Additional information of VOAs is suitable in the introduction mainly acids studied in this study.

-More information on sources and fate of studied VOAs was added to the introduction

5. The authors refer the role of acids in SOA formation, are the acids studied here relevant to SOA formation? Please elaborate?

- Studied VOAs participate into heterogeneous reactions on particles. This was clarified in the introduction.

6. It will be beneficial to clarify if gas phase or aerosol phase VOAs or both were measured here?

-This was clarified in the abstract, introduction and section 2.1

7. The discussion related to GC-MS sampling and analysis could be clarified. Confused about the sampling directly to the cold trap? Using test tubes etc.. How the

samples were taken from the field site!!

-We clarified this in section 2.1 and added references into the introduction on the use of in situ GC-MS for other compounds

6. The paper could be improved significantly if the sampling method was clearly stated. A schematic diagram may be beneficial?

-We clarified this in section 2.1. and added a schematic diagram of the sampling system as supplement Fig. S1.

7. Is the PTRTOFMS discussion necessary in this study?

-We prefer to have detailed description also on PTR-TOFMS, since we compare the results to the GC results and the PTR-TOFMS is not yet widely used instrument. We included the results of the PTR-TOF as it displays similar variations in the acetic and propionic acid concentrations. Furthermore it shows the reader that the PTR-MS also has problems measuring volatile organic acids and the value of using a reliable method for measuring VOAs in ambient conditions.

8. What the authors refer here "The second port of the three way valve was used for this." Page 5, line 2.

-This sentence was deleted

9. Please elaborate in "and their fragmentation pattern was not quantified" page 5, line 4

-When using PTR-MS most molecules (R) are charged by proton transfer, which adds an H+ to them. Certain oxygen containing compounds, like acetic acid and propanoic acid, can lose an OH during ionization. The ratio between those reactions depends on instrumental settings and affects the sensitivity of the compound directly. One way to characterize this ratio is by mixing pure acetic acid with VOC free air and measure it with the instrument. As we did not have pure acetic acid, we could not make this

fragmentation test and therefore estimated it by using literature values. We refrain from adding such a detailed description to the manuscript, but cited a study which explains contains this information. We rephrased the sentence to: The calibration gas did not contain acetic acid or propanoic acid, therefore the sensitivity was estimated. As both molecules fragment when measured with PTR-MS (von Hartungen et al., 2004), the sensitivities were estimated to be 50% of the acetone sensitivity.

10.How the memory effect can be incorporated in this study? It does depends on the acids concentrations in the previous runs? Please elaborate? How it was different from one acid to the other?

-There is some memory effect for all studied VOAs. Due to memory effects fast variations of mixing ratios would not be detected. However, this system is relatively slow (60 minute samples every other hour) and unable to detect fast variations anyway. Memory effect was clearly seen after running field standard gas which contained VOA, but it was only <3% already in the first run after the standard as shown by running blank gas ($N_2$ 6.0) after the field standard. We deleted always 5 samples after running this standard. In ambient samples variation of concentrations was much lower and therefore we expect that memory effect is not as big problem in ambient air samples as after standard gas. In inlet tests we did not detect any major losses of these compounds either. Discussion on memory effect was added to the text and figures of blanks and first runs after field standards were added to the supplementary material.

11. Confused about this statement "Using lower concentration for the calibrations would be expected to solve this issue." page 5, line 19? It did or did not solve the problem?

-Our standard gas for aromatic hydrocarbons, which were measured at the same time, contained high mixing ratios of VOAs as contaminant and it was not possible to lower the concentration in this case due to flow limitations and mixing ratios of other compounds in the standard. Nevertheless, as mentioned earlier more discussion on memory effect was added to the main text and two supplement Figures (S4 and S5) to clarify this.

12. Which N2 flow the authors refers here?? "The precision (Uprec) was checked by injecting known amounts of acids into the N2 flow."?

-This was clarified in section 3.1

---

## Author Comment (AC3) · 23 Nov 2016

Thank you for the good comments. We have carefully considered all the comments and improved our manuscript according to them. Please, find below our answers to the general and specific comments.

One of my major concern is the question how quantitative is the presented method for the target analytes. The authors report by themselves that especially acetic acid showed a problematic behavior and that large deviations were observed. Just considering the sampling step in the "cold" trap at 25 C (to avoid water interference) in combination with the calibration using liquid standards injected into adsorbent tubes(purged with nitrogen – again to avoid water interference) results in the inherent difficulty to estimate a reliable recovery, since for both steps losses of the analyte (e.g.esperically

acetic acid) cannot be excluded (or better have to be expected!?). If no additional experiments can be performed, at least a comprehensive discussion about such losses are needed, which would certainly improve the manuscript.

-We added some chromatograms and calibration curves as supplementary material to better justify the method. Cold trap was filled with hydrophobic adsorbent, water goes through the trap and organic compounds are retained on it. However, excess water can block the cold trap if temperature of the cold trap is below the dew point of the sampled air. Therefore we kept this hydrophobic cold trap at 25°C. We have seen already in earlier measurements (with e.g. monoterpenes) that if we use lower cold trap temperature during warm summer days, water can block the cold trap. Also calibration samples from the adsorbent tubes were directed to the cold trap at 25°C and before that adsorbent tubes (filled also with hydrophobic adsorbents) were purged with nitrogen to remove excess water used as solvent (1 $\mu$l injected into the tube). This was so small amount (1$\mu$l) that most probably this would not have been even needed. The main issue of the adsorbent tubes was high background values and not losses. We added to section 2.1 the explanation for water removal in our study.

Specific remarks:

Page 6: "Some memory effect was found". Please describe more quantitative these effects since especially for the highly polar and sticky analyte molecules such a behavior has to be known in detail.

-There is some memory effect for all studied VOAs. This was clearly seen after running field standard gas of benzene and toluene which contained VOAs as contaminant, but the memory effect was <3% already in the first blank run after the standard. We deleted always 5 samples after running this standard. In ambient samples variation of concentrations was much lower than differerence between standard gas and ambient and therefore we expect that memory effect is not as big problem in ambient air samples as after standard gas. In inlet tests we did not detect any major losses of these

compounds even with humidified samples. Discussion on memory effect was added to the text and figures of blanks and first runs after field standards were added to the supplementary material.

Page 8, line 20: As also discussed later in the manuscript: High concentrations during nighttime are not necessarily a consequence of nitrate chemistry (e.g. transport).

- It is true that the lower boundary layer present during the nigh may also explain higher nighttime mixing ratios, but since this clear peak was not found for any other compounds than for butanoic acid, we believe that there was an additional butanoic acid source during that night. An explanation for this was also added to the text in section 3.2.1.

Please check the whole text for typos (using a spell checker!?)

- This was done.